# P2X_7_ Receptor and Extracellular Vesicle Release

**DOI:** 10.3390/ijms24129805

**Published:** 2023-06-06

**Authors:** Maria Teresa Golia, Martina Gabrielli, Claudia Verderio

**Affiliations:** National Research Council of Italy, Institute of Neuroscience, Via Raoul Follereau 3, 20854 Vedano al Lambro, Italy; mariateresa.golia@in.cnr.it (M.T.G.); martina.gabrielli@in.cnr.it (M.G.)

**Keywords:** P2X_7_ receptor, extracellular vesicles, inflammation, neurodegeneration

## Abstract

Extensive evidence indicates that the activation of the P2X_7_ receptor (P2X_7_R), an ATP-gated ion channel highly expressed in immune and brain cells, is strictly associated with the release of extracellular vesicles. Through this process, P2X_7_R-expressing cells regulate non-classical protein secretion and transfer bioactive components to other cells, including misfolded proteins, participating in inflammatory and neurodegenerative diseases. In this review, we summarize and discuss the studies addressing the impact of P2X_7_R activation on extracellular vesicle release and their activities.

## 1. Introduction

P2X_7_ receptor (P2X_7_R) is an ATP-gated ion channel belonging to the purinergic P2X family. It is highly expressed by cells of the innate immune system, especially macrophages [1], dendritic cells [2], mast cells [3], and microglia [4], where it promotes inflammasome formation and the release of inflammatory cytokines [5,6,7]. P2X_7_R is also present in adaptive immune cells (T cells), where it regulates cell development and function [8], and in many other cell types [9] including nervous system cells [10,11], epithelial and endothelial cells [12,13], bone cells [14], fibroblasts [15], and smooth muscle cells [16], as well as in tumor cells, where its expression often correlates with a worse diagnosis [17,18].

Among the P2XR family, P2X_7_R exhibits peculiar features including a low affinity for ATP, a lack of desensitization and unique structural domains, i.e., a “C-cysteine anchor” intra-cytoplasmic motif and a long C-terminal cytoplasmic domain that contains several protein–protein interaction motifs [19,20]. The receptor also exhibits a characteristic dual gating state depending on extracellular ATP (eATP) concentration. At micromolar eATP concentration, P2X_7_R opens a cation-selective channel that mediates the cellular influx of Na^+^ and Ca^2+^ ions and an efflux of K^+^ [21]; at higher eATP concentrations (above 100 mM) and upon prolonged exposure, the receptor functions as a non-selective membrane pore permeable to hydrophilic molecules [21], generally leading to cytotoxicity and apoptotic cell death [22]. Ion alterations also induce the opening of the pannexin-1 channel, which, by releasing ATP into the extracellular space, perpetuates P2X_7_R stimulation [23]. Therefore, whether pannexin-1, as a P2X_7_R-associated protein, forms the large pore itself or mediates its formation is still a matter of debate.

Channel opening increases cell proliferation and survival [24,25] whereas large pore opening induces the activation of inflammasome (as reviewed in [26]), a cytoplasmic multiprotein complex that, in response to pathogens/cell damage, triggers cytokine release and pyroptosis, a lytic form of programmed cell death [27].

The inflammasome consists of a sensor protein (e.g., NLR family CARD domain containing 4 (NLRC4), NLR Family Pyrin Domain Containing 1 (NLRP1), NLR Family Pyrin Domain Containing 3 (NLRP3) that is activated by ATP, absent-in-melanoma 2 (AIM2), and pyrin), an inflammatory caspase, and in some cases an adaptor protein, such as ASC (apoptosis-associated speck-like protein containing a CARD) [28]. Once assembled and activated in response to ATP, the NLRP3 inflammasome triggers pro-caspase-1 cleavage, which generates active caspase-1 that, in turn, drives the enzymatic activation of the leaderless cytokines Interleukin (IL)-1β and IL-18, initiating an inflammatory response [29,30].

In addition to ATP, the P2X_7_R cation channel can be opened by non-ATP nucleotides, such as NAD^+^ (nicotinamide adenine dinucleotide). This ATP-independent pathway consists of receptor ADP-ribosylation by ADP-ribosyltransferases ART2.1 and ART2.2, which catalyze the transfer of ribose from NAD^+^ to arginine 125 in the ectodomain of the P2X_7_R close to the ATP binding site [31]. The P2X_7_R opening by ADP-ribosylation enables Ca^2+^ and Na^+^ influx and K^+^ efflux, phosphatidylserine externalization, membrane pore formation, mitochondrial membrane breakdown, and ultimately cell death [32,33,34].

Interestingly, both ATP and NAD^+^ concentrations are low (in the submicromolar range) in the extracellular space, due to the activities of the ectoenzymes CD39 and CD38 that degrade them, respectively [8,35]. Therefore, P2X_7_R activation occurs at inflammatory or damaged sites, as well as in the tumor microenvironment, where ATP and NAD^+^ are released in substantial amounts [36,37]. Accordingly, P2X_7_R-mediated signaling is activated in a large variety of Central Nervous System (CNS) disorders (i.e., Alzheimer’s, Parkinson’s and Huntington’s disease, multiple sclerosis, Amyotrophic Lateral Sclerosis, stroke, neurotrauma, neuropathic pain, epilepsy, and neuropsychiatric disorders), and P2X_7_R antagonists are under intense investigation as a therapy for these conditions (as reviewed in [38]).

One of the main consequences of P2X_7_R activation is the formation of blebs at the cell surface and the release of extracellular vesicle (EVs) into the microenvironment. EVs are a heterogeneous group of cell-derived membranous structures which directly bud from the plasma membrane (microvesicles) or originate in the endocytic compartment as intraluminal vesicles (ILVs) which are released through the fusion of multivesicular bodies (MVBs) to the plasma membrane (exosomes) [39]. Due to technical limitations in isolating and distinguishing EVs based on their biogenesis, the currently recognized nomenclature identifies EVs according to their physical properties and dimensions, distinguishing medium-large/large EVs (>200 nm) and small EVs (<200 nm) [40]. Accordingly, here we use the terms large and small EVs to refer to the two main populations of EVs. EVs act as carriers of bioactive molecules (proteins, lipids, genetic materials, and metabolites) and convey their bioactive cargoes between cells, playing a fundamental role in cell-to-cell communication in both physiological conditions and during inflammatory and degenerative diseases [41,42].

In the present review, we will first discuss the impact of P2X_7_R activation on EV release from the cell surface and the endocytic compartment. Then, we will summarize the current knowledge about the role of P2X_7_R activation in the sorting of proteins into EVs, the secretion of inflammatory cytokines, and the dissemination of misfolded proteins. 

## 2. P2X_7_R Activation and EV Release

Among stimuli that promote EV release (cytokines, LPS, capsaicin, serotonin, Wnt3a, and a-synuclein) [43], eATP is the classical trigger that, through P2X_7_R activation, massively increases the shedding of EVs from the plasma membrane of immune cells including dendritic cells [44,45], microglia [46], and macrophages [6,47]. Of note, not only millimolar concentration of eATP but also ATP endogenously released by astrocytes could induce P2X_7_R-dependent EV release in microglia-astrocyte co-cultures [46].

The first evidence implicating P2X_7_R activation in the release of EVs dates back to 2001, when MacKenzie and colleagues showed that within the first few minutes of P2X_7_R activation, bleb formation occurs at the surface of monocytes and large EVs with externalized phosphatidylserine (PS) are released into the extracellular space as a result of bleb detachment from the membrane [48]. Notably, bleb formation and the externalization of PS, a typical marker of apoptosis, are reversible processes under brief P2X_7_R stimulation, dissociating ATP-induced bleb formation and EV release from apoptosis [48].

Subsequent studies have clarified the mechanism by which P2X_7_R activation drives large EV biogenesis. The pathway involves the activation of p38 MAP kinase and Rho-associated protein kinases (ROCK) [49,50,51,52] (Figure 1). Specifically, following P2X_7_R activation, a Src kinase interacts with the C-terminus of the receptor and phosphorylates p38 MAP kinase, inducing the translocation of acid sphingomyelinase (A-SMase) from the luminal lysosomal compartment to the plasma membrane outer leaflet [49]. This enzyme hydrolyzes sphingomyelin to ceramide, thereby creating ceramide-enriched microdomains that, by perturbing membrane curvature/fluidity, facilitate the formation of plasma membrane blebs and large EV shedding from microglia and astrocytes [49]. The key role of A-SMase in EV release was indicated by the genetic inactivation and pharmacological inhibition of the A-SMase. Both approaches strongly abolished EV release from LPS-primed microglia and astrocytes [49] and alveolar macrophages [7] (see below). The in vivo validation of the role of P2X_7_R and A-SMase in EV release was suggested by the immunohistochemical quantification of EV-like particles immunoreactive for the P2X_7_R in the cerebral cortex of rats administered intraperitoneally with the P2X_7_R antagonist A804598 or the A-SMase inhibitor FTY720, immediately after traumatic brain injury (TBI), a condition inducing P2X_7_R expression and EV release from microglia [53]. Both drugs reduced the number of P2X_7_R positive particles surrounding microglia, but the particles were not unequivocally identified as EVs [53]. Beyond lipids, the P2X_7_R-dependent activation of ROCK is another key intracellular signaling mechanism in plasma membrane bleb formation. By interacting with LIM-kinases (LIMK) and the myosin light chain (MLC) [54,55], ROCK regulates cytoskeletal reorganization, favoring the shedding of large EV. The role of ROCK in EV release was supported by the complete abrogation of P2X_7_R-dependent blebbing in HEK293 cells pre-incubated with ROCK inhibitor Y-27632 [50].

In addition to large EVs shed from the plasma membrane, P2X_7_R activation promotes the release of small EVs generated in the endosomal compartment of innate immune cells [56,57,58]. Interestingly, in human macrophages the ATP-dependent small vesicles’ release was shown to be a consequence of NLRP3/ASC/procaspase-1 inflammasome assembling, as evidenced by the suppression of small EVs secretion under the genetic deletion of ASC or NLPR3 [57]. These findings suggest that inflammasome activation may regulate the membrane trafficking pathways that control MVBs fusion to the plasma membrane. The involvement of the NLRP3 inflammasome in small EV secretion was further indicated by a study showing that LPS/ATP-induced inflammasome and caspase-1 activation promotes the loading of specific miRNAs into small EVs and their release via the cleavage of the Rab-interacting lysosomal protein RILP in a human monocytic cell line. RILP is part of the complex that links the trafficking GTPase Rab7 to the dynein motor complex; cleaved RILP does not make the link to the dynein complex and promotes the movement of MVBs toward the cell surface (Figure 1). In addition, it induces selective miRNA cargo sorting via interaction with Hrs (hepatocyte growth factor-regulated tyrosine kinase substrate), a component of the ESCRT-0 complex, and the RNA-binding protein FMRP that acts as a chaperone to package specific AAUGC motif-containing miRNAs into ILV [59]. Accordingly, the inhibition of caspase-1 blocked small EV secretion from the monocytic cells activated with LPS/ATP [59]. Further advances in the mechanism driving the ATP-induced release of small EVs were made in 2022, when Ruan and colleagues identified Sepp1, Mcfd2, and Sdc1 as critical molecules for the release of CD63 positive small EVs from microglia using a genome-wide shRNA library screening [60]. The identified molecules may represent interesting targets for inhibiting small EV release and limiting the pathogenic contribution of EVs and their inflammatory cargo to neuroinflammatory disorders.

## 3. Role of P2X_7_R-Induced EVs in Cytokine Release and the Propagation of Inflammation

Cytokines lacking the conventional secretory sequence do not follow the classical endoplasmic-reticulum-to-Golgi pathway for secretion, but are exported via membrane-enclosed vesicles [61].

MacKenzie and colleagues showed that the pro-inflammatory cytokine IL-1β is packaged in EVs shed upon P2X_7_R-mediated monocyte activation, providing the first evidence that P2X_7_R-induced EV release represents an unconventional mechanism for the secretion of a leaderless protein [48]. In the following years, the presence of IL-1β was confirmed in EVs released from rat microglia and human dendritic cells [45,46], and the way in which IL-1β passes from the EVs lumen into the extracellular space was clarified. Specifically, it was observed that EVs contain the machinery necessary for IL-1β processing (they carry P2X_7_R in their membranes and caspase-1 in their lumen), and that P2X_7_R opening at the EV surface activates caspase-1-dependent IL-1β cleavage in the EVs, similar to the cells [45,46] (Figure 2). In addition, evidence was provided that IL-1β release may occur through the EV membrane as a consequence of P2X_7_R-dependent pore opening [5], and that pannexin-1 is required for IL-1β processing and secretion from macrophages [23].

Collectively, these results indicated that large EVs released upon P2X_7_R activation from immune cells carry IL-1β and mediate IL-1β secretion in a P2X_7_R-dependent manner. Later evidence obtained from macrophages and dendritic cells showed that small EVs also carry inflammasome components, i.e., NLP3, caspase-1, and ASC, that are essential for IL-1β processing within EVs [57,58,62]. These studies also showed that both small and large EVs released from mycobacterium-infected macrophages and dendritic cells upon P2X_7_R activation are enriched in major histocompatibility complex class II (MHC-II) [45,63,64], thus potentially contributing to the rapid dissemination and presentation of foreign antigens as part of the immune response induced by local inflammation [65,66]. In line with the involvement of EVs in the immune response, large EVs released upon P2X_7_R activation from LPS-treated microglia were reported to carry the IL-1β transcript and to act as vehicles for the transfer of IL-1β mRNA between immune cells, participating in the propagation of inflammatory signals both in vitro and in vivo, upon EV injection into the mouse corpus callosum [67].

Subsequent studies have revealed that large EVs released upon P2X_7_R activation mediate the release of other inflammatory cytokines, i.e., IL-18 and Tumor necrosis factor (TNF) [6,7] (Figure 2). Like IL-1β, IL-18 is a leaderless cytokine and its release from human blood-derived macrophages occurs in association with large EVs generated upon P2X_7_R activation [6]. Conversely, TNF is secreted by the classical endoplasmic-reticulum-to-Golgi pathway in a mature, soluble isoform of 17 kDa. Thus, the presence of TNF in EVs was quite unexpected. Notably, Soni and colleagues demonstrated that ATP stimulation alters the mechanism of TNF secretion from mouse bone-marrow derived macrophages, redirecting TNF release from classical to unconventional secretory pathways [7]. Specifically, ATP inhibits the conventional secretion of soluble TNF and drives the packaging of the transmembrane pro-TNF isoform into large EVs [7]. TNF carried by EVs was biologically more potent than soluble TNF at equal or even higher doses and mediated significant lung inflammation in vivo [7], revealing that ATP-dependent packaging into EVs uniquely protects enclosed TNF, enhancing its biological activity. These findings were confirmed in vivo upon the intratracheal instillation of ATP and the analysis of EV production and TNF quantification in the bronchoalveolar lavage fluid [7].

To conclude, relevant cytokines are expressed in EVs at both mRNA and protein levels and in both transmembrane/pro- and soluble/mature forms. The cytokines can be rapidly released from vesicles in the mature forms (IL-1β and IL-18) at sites of extracellular ATP accumulation via P2X_7_R opening, or be presented to recipient cells (pro-TNF), promoting acute inflammation. Given that packaging into EVs prevents the degradation and dilution of the inflammatory mediators, cytokines-loaded EVs released by P2X_7_R-expressing cells can propagate long-distance inflammatory signals to recipient cells and tissues. Cytokines released as part of EVs upon P2X_7_R activation are listed in Table 1.

## 4. P2X_7_R Activation Influences the Proteome of EVs

Distinct EV populations are released by cells in response to various stimuli that influence the cellular activation state [43], with EV composition often mirroring that of donor cells. As already mentioned, P2X_7_R activation influences the miRNA selectivity of small EV cargo loading through interactions with the RNA-binding protein FMR1 [59]. Furthermore, a few studies have identified proteins that are released as part of EVs via a P2X_7_R-dependent mechanism (Table 1). These molecules seem to share the ability to control the inflammatory response.

Takenouchi and coworkers showed that only under P2X_7_R activation was glyceraldehyde-3-phosphate dehydrogenase (GAPDH), a key glycolytic enzyme and a leaderless cytoplasmic protein, sorted in small and large EVs released by LPS-treated microglial cells [69]. Once in the extracellular space, GAPDH might be involved in the regulation of neuroinflammation by favoring the phosphorylation of p38 MAP kinase in microglia [69]. CD14 is another abundant protein cargo of EVs released upon P2X_7_R activation from macrophages [68]. P2X_7_R-induced CD14 release in EVs ensures the maintenance of elevated concentrations of circulating CD14 which, by acting as a co-receptor for LPS, is fundamental to controlling infection and increasing survival during sepsis [68].

Further studies associated P2X_7_R activation with the release of proteins modulating or amplifying the inflammatory response. The release of Interleukin-1 receptor antagonist (IL-Ra) occurs via a P2X_7_R-dependent large EV-shedding mechanism in macrophages [70]. Since IL-Ra functionally inhibits IL-1-dependent cellular activation, maintaining a balance between IL-1 and IL-1ra may be an important mechanism for regulating the overall inflammatory response [81]. Conversely, the mature form of the TNF converting enzyme (TACE), which is part of the small EVs produced by LPS-primed macrophages under P2X_7_R activation, by processing the membrane-bound TNF into a soluble cytokine may amplify the pro-inflammatory responses [71]. Finally, other studies have linked P2X_7_R activation to an increased release of tissue factor (TF) containing large EVs from human dendritic cells and macrophages, producing an enhanced pro-thrombotic response [44,72].

To the best of our knowledge, only one label-free proteomic study systematically explored how P2X_7_R activation influences the protein composition of EVs. This work showed that the proteome composition of EVs (large and small) released from microglia under ATP activation is largely distinct from that of constitutively released EVs [56]. Specifically, it revealed that EVs released under ATP stimulation contain an increased number of proteins involved in antigen processing and presentation which, along with inflammatory cytokines and MHC-II (see chapter above), can participate in the immune response. In addition, EVs released under P2X_7_R activation show an increased number of autophagy-lysosomal proteins (i.e., Cathepsin D and C, Lamp1, Vcp, and CD68), suggesting an enhancement of the degradative pathways, and are enriched in proteins implicated in adhesion/extracellular matrix organization (Fibulin 1, Comp, Plasminogen and the matricellular proteins thrombospondin 1 and 4, Vinculin, and Fermt3), which likely account for the stronger EV adhesion to astrocytic target cells compared to constitutive EVs. Interestingly, ATP also drives the sorting in EVs of a set of proteins involved in energy metabolism (i.e., Gpi, Ldha, Mdh2, Tranketolase, Glutamate dehydrogenase 1, Acacb, and others), which together with the glycolytic enzyme GAPDH identified by Takenouchi and colleagues may influence the metabolism of receiving cells. Finally, a unique set of cytoskeletal proteins and proteins regulating the dynamics of actin filaments have been detected in EVs released upon P2X_7_R activation, i.e., the capping actin protein Capzb, Cap1, and ARP2 actin related protein. These proteins, by controlling the organization of actin filaments present in a fraction of large EVs [82], may favor changes in the EV morphology and promote the capacity of a small fraction of glial EVs to actively move at the surface of target cells [73,82]. Interestingly, some of these cytoskeletal proteins interact with the C-terminus of the P2X_7_R [83], thus supporting a direct role for the receptor in the sorting of the protein cargo.

Further studies are necessary to define whether changes occurring in microglia-derived EVs under P2X_7_R activation may be common to EVs released by other cells following receptor stimulation.

## 5. P2X_7_R Activation and Misfolded Protein Release in EVs: Implications in Neurodegeneration 

Among the bioactive cargo of EVs released upon P2X_7_R activation, there are pathological misfolded proteins including beta amyloid (Aβ) [73,74,75], tau protein [76,77,78], and α-synuclein [79,80] (for an extensive review, see [43] and Table 1).

By spreading throughout the brain in association with EVs, Aβ and tau protein contribute to the progression of neurodegeneration in AD and tauopathies (reviewed in [84]). Specifically, it has been demonstrated that EV-associated tau released by microglia after ATP stimulation, but not an equal amount of free tau, are able to mediate efficient tau propagation in the mouse hippocampus [76]. The pivotal involvement of P2X_7_R in this process has been proven by recent findings showing that the administration of the orally applicable and CNS-penetrant P2X_7_R selective antagonist GSK1482160, which inhibits EV secretion from microglia, blocks tau propagation and rescues memory impairment in the P301S mouse model of tauopathy [77]. Furthermore, the suppression of tau accumulation in the hippocampal region has been indicated in P301S mice lacking P2X_7_R (P2X_7_R^−/−^:P301S mice) [85]. Although for the EV-mediated propagation of Aβ no direct proof of P2X_7_R involvement by in vivo inhibition/depletion is currently available, large EVs released upon ATP activation by Aβ-exposed microglia, and injected in the mouse brain parenchyma, were shown to cause amyloid-related impairment of synaptic plasticity and propagate deficits to synaptically connected regions [73]. Again, free oligomeric Aβ was not able to propagate synaptic alterations [73]. 

Small EVs released upon ATP stimulation can also transfer α-synuclein, a key molecule in Parkinson’s disease pathogenesis, from microglia to neurons, where they act as seeds to aggregate the native protein [79]. Once injected in the striatum of healthy mice, microglial small EVs carrying α-synuclein, but not free α-synuclein, cause the aggregation of the protein at the injection site and in anatomically connected regions, and the loss of dopaminergic neurons in the nigrostriatal pathway associated with movement disorders months later [80].

Interestingly, P2X_7_R’s expression and function have been found to be altered in both AD/tauopathies patients and mouse models, especially in microglia and astrocytes surrounding amyloid plaques, while its genetic or pharmacological inhibition ameliorated the pathology in mice, mitigating inflammation and improving cognitive defects [86,87,88,89]. For these reasons, P2X_7_R has been implicated in both Aβ and tau-mediated neurodegeneration [87,89] and recognized as a promising pharmacological target for AD [90]. This also applies to Parkinson’s disease, as the pharmacological inhibition of P2X_7_R signaling limited hemi-Parkinsonism symptoms in a rat model of 6-hydroxydopamine-induced nigrostriatal lesion [91,92,93]. The involvement of the receptor in the EV-mediated propagation of misfolded proteins strengthened its potential as a therapeutic target for neurodegenerative diseases.

The presence of misfolded proteins in EVs released upon P2X_7_R activation indicates that EV release represents a mechanism exploited by cells to dispose of toxic material, which cannot be degraded in the cells, an old hypothesis formulated many years ago when EVs were still considered cellular debris or culture artefacts, and currently supported by many findings [73,75,79,94].

## 6. Conclusions

At inflammatory or damaged sites, P2X_7_R activation by extracellular ATP or NAD^+^ promotes the massive shedding of large EVs from the plasma membrane, via the translocation of acid sphingomyelinase and the release of small EVs from multivesicular bodies via inflammasome activation. The generated EVs expose MHCII on their surface and specific inflammatory miRNAs cargo in their lumen, and carry and release inflammatory cytokines into the extracellular space, promoting a local acute inflammatory response. Encapsulation into EVs can enhance cytokine activity, as shown for TNF, and by preventing cytokine degradation can deliver inflammatory signals to distant cells and tissues. 

P2X_7_R-dependent EV release also represents a mechanism for the cells to dispose of unwanted materials, such as misfolded proteins (aβ, tau, and a-synuclein), which are resistant to degradation, and to disseminate them throughout the brain. Encapsulation into EVs can also increase the activity of misfolded proteins. In fact, Aβ, tau, and α–synuclein induce/propagate pathology more efficiently when associated with EVs, indicating that a higher activity of EV-associated proteins compared to free soluble forms is not a mere consequence of protection from degradation. 

Further research is needed to better characterize the molecules modulating or amplifying the inflammatory/degenerative responses that are released as part of EVs upon P2X_7_R activation, in light of the emerging role of P2X_7_R inhibitors as promising therapeutic tools for limiting neurodegenerative and inflammatory processes.

## Figures and Tables

**Figure 1 ijms-24-09805-f001:**
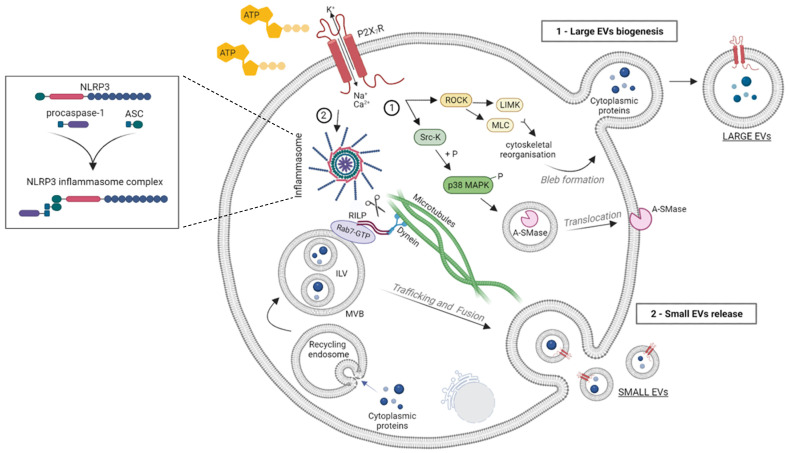
Schematic representation of EV release upon ATP-induced P2X_7_R activation. Upon ATP stimulation, P2X_7_R activation induces the recruitment of an Src kinase at the C-terminus of the receptor and the activation of ROCK and p38 MAP kinase triggering cytoskeletal reorganization (i.e., MLC and LIMK) and the translocation of A-SMase to the outer leaflet of the plasma membrane. A-SMase hydrolyzes sphingomyelin to ceramide, facilitating blebs’ formation and large EV shedding (route 1). P2X_7_R activation and the consequent efflux of K^+^ also promotes the release of small EVs, generated in the endosomal compartment as ILVs, by inducing NLRP3/ASC/procaspase-1 inflammasome assembling. Inflammasome activation regulates the membrane trafficking pathways that control MVB fusion to the plasma membrane via the cleavage of RILP (route 2). Image created with BioRender.com (accessed on 15 March 2023).

**Figure 2 ijms-24-09805-f002:**
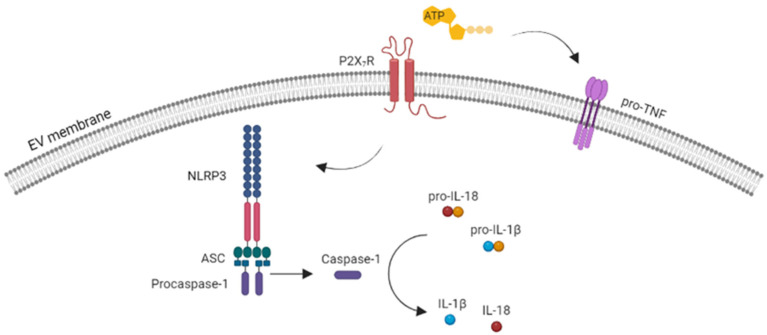
Schematic representation of cytokine processing in EVs upon ATP-induced P2X_7_R activation. P2X_7_R activation, by inducing NLRP3 inflammasome assembling, triggers pro-caspase-1 cleavage, which generates active caspase-1 that, in turn, drives the enzymatic activation of the leaderless pro-inflammatory cytokines IL-1β and IL-18. On the other hand, ATP induces TNF packaging into EVs as a transmembrane pro-TNF isoform. Image created with BioRender.com (accessed on 15 March 2023).

**Table 1 ijms-24-09805-t001:** Proteins released as part of EV cargo upon P2X_7_R activation and their phato/physiological impact.

EV Cargo	EV Type	EV Cellular Source	Involved Patho/Physiological Processes	Refs.
Fibulin 1, Comp, Plasminogen and the matricellular proteins thrombospondin 1 and 4, Vinculin, and Fermt3	Small and large EVs	Rat microglia	Adhesion/extracellular matrix organization	[56]
Cathepsin D and C, Lamp1, Vcp, and CD68	Small and large EVs	Rat microglia	Autophagy-lysosomal pathway	[56]
Capzb, Cap1, and ARP2 actin-related protein	Small and large EVs	Rat microglia	Cytoskeleton organization	[56]
MHC-II	Small EVs	Murine macrophages and dendritic cells	Dissemination and presentation of foreign antigens	[63]
	Large EVs	Human dendritic cells		[45]
	Small and large EVs	Murine macrophages		[64]
Gpi, Ldha, Mdh2, Tranketolase, Glutamate dehydrogenase 1, Acacb, and others	Small and large EVs	Rat microglia	Energy metabolism	[56]
CD14	EVs	Murine macrophages	Inflammation	[68]
GAPDH	Small and large EVs	Murine microglia		[69]
IL-18	Large EVs	Human macrophages		[6]
IL1β	Large EVs	Human monocytes Rat microglia Human dendritic cells		[48] [46] [45]
	Small EVs	Murine macrophages		[57]
IL-Ra	Large EVs	Murine macrophages		[70]
Inflammasome components	Large EVs	Rat microglia Human dendritic cells		[46] [45]
	Small EVs	Murine macrophages Murine microglia		[57] [62]
TACE	Small EVs	Mouse macrophages		[71]
TF	Large EVs	Human dendritic cells Murine macrophages		[44] [72]
TNF	Large EVs	Murine macrophage		[7]
Aβ	Large EVs	Murine microglia Rat microglia	Neurodegeneration	[73,74] [75]
Tau protein	Small EVs	Murine microglia		[76,77]
	Small EVs	Murine microglia		[78]
α-synuclein	Small EVs	Murine microglia		[79,80]

## Data Availability

Not applicable.

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
