# Peer review of "P2X7 Receptor and Extracellular Vesicle Release"

_ijms, 2023, doi:10.3390/ijms24129805_

Round 1

Reviewer 1 Report

Activation of P2X7 receptor (P2X7R) is strictly associated with the release of extracellular vesicles. This review summarized the studies addressing the impact of P2X7R activation on extracellular vesicle release and their activities.

1.          The molecular regulation between P2X7 receptor and extracellular vesicle release were not present in this manuscript. Clearly describing these molecular pathways makes the article more complete.

2.          The application of P2X7R modulation on neuro-diseases or disorders may clear described in the manuscript.

3.          Figures may help the readers to understand the pathway described in the paragraph “3. Role of P2X7R-induced EVs in cytokine release and propagation of inflammation”.

4.          There are type errors in the text. For example, p38 MAP kinase and p38 Map kinase. Authors have to check the manuscript carefully.

Minor editing of English language required.

Reviewer 2 Report

This review covers the role of ATP and NAD+ sensitive P2X7 receptors in the release of extracellular vesicles (EVs), mainly in the context of glial or immune cells.  The review discusses P2X7’s role in the activating intracellular pathways responsible for the release of both small EVs (from multivesicular bodies) and large EVs (shed from the plasma membrane), as well as research outlining P2X7’s role in determining EV cargo (e.g., cytokines, nucleic acids and proteins) and how that cargo mediates cellular function/dysfunction.  The review is well-written and acceptably comprehensive; I only have a few comments.

 Comments/Concerns

1It is hard to discuss the P2X7 receptor without at least mentioning pannexin channels.  Pannexins are ATP permeable channels that are often coupled with P2X7 channels, allowing eATP levels to rise to the high levels locally required to activate P2X7 channels.  Due to this coupling, and to the fact that P2X7 agonists often activate P2X7 channels, it can be hard to differentiate the activity of pannexin channels vs P2X7 channels.  Hence there is some debate in the literature on whether the large pore formed by eATP is P2X7 channels or pannexin channels (see Pelegrin and Surprenant, EMBO J, 25: 5071-82, 2006., for example).  This should be at least mentioned in the text. 

2There is at least one instance of the authors citing a review article in place of a primary research article (line 51, reference to # 31, referring to NAD+ activation of P2X7).  The authors should make sure to reference primary literature, and not reviews when discussing specific experimental results.

3The final 3 paragraphs (on page 8) should be under the heading “conclusions” or “summary”.

Reviewer 3 Report

The authors of the manuscript entitled “P2X7 RECEPTOR AND EXTRACELLULAR VESICLE RELEASE” have described the relevance of P2X7 activation with the release of extracellular vesicle. The detailed mechanisms of EV release  and propagation of inflammation are described. The review is well organized and easy to follow with proper citations. I find this review very well written and I did not identify any major concerns limiting the publishing of the manuscript as it is.  

Author Response

We thank the Reviewer for the positive comments.